# AdaProj: Adaptively Scaled Angular Margin Subspace Projections for Anomaly Detection with Auxiliary Classification Tasks

## Abstract

One of the state-of-the-art approaches for semi-supervised anomaly detection is to first learn an embedding space and then estimate the distribution of normal data. This can be done by using one-class losses or by using auxiliary classification tasks based on meta information or self-supervised learning. Angular margin losses are a popular training objective because they increase intra-class similarity and avoid learning trivial solutions by reducing inter-class similarity. In this work, AdaProj a novel loss function that generalizes upon angular margin losses is presented. In contrast to angular margin losses, which project data of each class as close as possible to their corresponding class centers, AdaProj learns to project data onto class-specific subspaces. By doing so, the resulting distributions of embeddings belonging to normal data are not required to be as restrictive as other loss functions allowing a more detailed view on the data. This enables a system to more accurately detect anomalous samples during testing. In experiments conducted on the DCASE2022 and DCASE2023 datasets, it is shown that using AdaProj to learn an embedding space significantly outperforms other commonly used loss functions achieving a new state-of-the-art performance on the DCASE2023 dataset.

## 1 Introduction

Semi-supervised anomaly detection is the task of training a system to differentiate between normal and anomalous data using only normal training samples (Aggarwal, 2017). For many applications, it is much less costly to collect normal data than anomalous samples for training a system because anomalies occur only rarely and intentionally generating them is often costly. Therefore, for these applications a semi-supervised anomaly detection setting is often a realistic assumption. An example application is acoustic machine condition monitoring for predictive maintenance, which is largely promoted through the anomalous sound detection (ASD) tasks of the annual DCASE Challenge (Koizumi et al., 2020; Kawaguchi et al., 2021; Dohi et al., 2022a; 2023) and will serve as the running example in this work. Here, normal data corresponds to sounds of fully functioning machines whereas anomalous sounds indicate mechanical failure. The goal is to develop a system that detects anomalous recordings using only normal recordings as training data.

One of the main difficulties to overcome is that it is practically impossible to record isolated sounds of a target machine. Instead, recordings also contain many other sounds emitted by non-target machines or humans. Compared to this complex acoustic scene, anomalous signal components of the target machines are very subtle and hard to detect without utilizing additional knowledge. Another main difficulty is that a system should also be able to reliably detect anomalous sounds when changing the acoustic conditions or machine settings without needing to collect large amounts of data in this changed conditions or to re-train the system (domain generalization (Wang et al., 2021)). One possibility to simultaneously overcome both difficulties is to learn mapping audio signals into a fixed-dimensional vector space, in which representations belonging to normal and anomalous data, called embeddings, can be easily separated. Then, by estimating the distribution in the embedding space of normal training samples, one can compute an anomaly score for an unseen test sample by computing the likelihood of this sample being normal or simply measuring the distance to normal samples. To train such an embedding model, the state-of-the-art is to utilize

an auxiliary classification task using provided meta information or self-supervised learning (SSL). This enables the embedding model to closely monitor target signals and ignore other signals and noise (Wilkinghoff & Kurth, 2023). For machine condition monitoring, possible auxiliary tasks are classifying between machine types (Giri et al., 2020; Lopez et al., 2020; Inoue et al., 2020) or, additionally, between different machine states and noise settings (Venkatesh et al., 2022; Nishida et al., 2022; Deng et al., 2022), recognizing augmented and non-augmented versions of normal data (Giri et al., 2020; Chen et al., 2023) or predicting the activity of machines (Nishida et al., 2022). Using an auxiliary task to learn embeddings is also called outlier exposure (OE) (Hendrycks et al., 2019) because normal samples belonging to other classes than a target class can be considered proxy outliers (Primus et al., 2020).

The contributions of this work are the following. First and foremost, AdaProj, a novel angular margin loss function that learns class-specific subspaces for training an embedding model, is presented. Second, it is proven that AdaProj has arbitrarily large optimal solution spaces allowing to relax the compactness requirements of the class-specific distributions in the embedding space. Last but not least, AdaProj is compared to other commonly used loss functions. In experiments conducted on the DCASE2022 and DCASE2023 ASD datasets it is shown that AdaProj outperforms all other loss functions. As a result, a new state-of-the-art performance is achieved on the DCASE2023 dataset[1].

## 1.1 RELATED WORK

When training a neural network to solve a classification task, usually the softmax function in combination with the categorical cross-entropy (CCE) is used. However, directly training a network this way only reduces inter-class similarity without explicitly reducing intra-class similarity (Wang et al., 2018). When training an embedding model for anomaly detection, high intra-class similarity is a desired property to cluster normal data and be able to detect anomalous samples. To address this issue, losses should also explicitly increase intra-class similarity.

There are several loss functions to achieve this. Ruff et al. (2018) proposed a compactness loss to project the data into a hypersphere of minimal volume for one-class classification. However, for machine condition monitoring in noisy conditions it is known that one-class losses perform worse than losses that also discriminatively solve an auxiliary classification task (Wilkinghoff & Kurth, 2023). Perera & Patel (2019) did this by simultaneously using a so-called descriptiveness loss consisting of a CCE on another arbitrary dataset than the target dataset, to be able to learn a better structured embedding space in case no meta information are available on the target dataset. For machine condition monitoring, often meta information is available as it can at least be ensured which machine is being recorded when collecting data. Inoue et al. (2020) used center loss (Wen et al., 2016), which minimizes the distance to learned class centers for each class. Another choice are angular margin losses that learn an embedding space on the unit sphere while ensuring a margin between classes leading to better generalization capabilities than losses that utilize the whole Euclidean space. Specific examples are the additive margin softmax loss (Wang et al., 2018) as used by Lopez et al. (2020; 2021) and ArcFace Deng et al. (2019) as used by Giri et al. (2020); Kuroyanagi et al. (2021); Deng et al. (2022). Wilkinghoff (2021; 2023a) use the AdaCos loss (Zhang et al., 2019), which essentially is ArcFace with an adaptive scale parameter, or the sub-cluster AdaCos loss (Wilkinghoff, 2021), which utilizes multiple sub-clusters instead of a single one.

As stated above, the goal of this work is to further extend these loss functions to learning class-specific linear subspaces to relax the compactness requirements and allow more flexibility for the network when learning to map audio data into an embedding space. There are also other works utilizing losses to learn subspaces based on orthogonal projections to learn embedding spaces for other applications in different ways. Yu et al. (2021) used orthogonal projections as a constraint for training an autoencoder based anomaly detection system. Another example is semi-supervised image classification by using a combination of class-specific subspace projections with a reconstructions loss and ensure that they are different by also using a discriminative loss (Li et al., 2022). Our work focuses on learning an embedding space through an auxiliary classification task that is well-suited for semi-supervised anomaly detection.

---

[1]The source code will be made available after the review process to not reveal the identity of the authors.

## 2 METHODOLOGY

### 2.1 NOTATION

Let $\phi : X \rightarrow \mathbb{R}^D$ denote a neural network where $X$ denotes some input space, which consists of audio signals in this work, and $D \in \mathbb{N}$ denotes the dimension of the embedding space. Define the linear projection of $x \in \mathbb{R}^D$ onto the subspace $\text{span}(\mathcal{C}_k) \subset \mathbb{R}^D$ as $P_{\text{span}(\mathcal{C}_k)}(x) := \sum_{c_k \in \mathcal{C}_k} \langle x, c_k \rangle c_k$. Furthermore, let $\mathcal{S}^{D-1} = \{y \in \mathbb{R}^D : \|y\|_2 = 1\} \subset \mathbb{R}^D$ denote the $D$-sphere and define $P_{\mathcal{S}^{D-1}}(x) := \frac{x}{\|x\|_2} \in \mathcal{S}^{D-1}$ to be the projection onto the $D$-sphere.

### 2.2 ADAPROJ LOSS FUNCTION

Similar to the sub-cluster AdaCos loss (Wilkinghoff, 2021), the idea of the AdaProj loss is to enlarge the space of optimal solutions to allow the network to learn less restrictive distributions of the normal samples. This may help to differentiate between normal and anomalous data after training. The reason is that for some auxiliary classes a strong compactness may be detrimental when aiming to learn an embedding space that separates normal and anomalous data since both may be mapped onto the same compact distribution making it impossible to distinguish them. This relaxation is achieved by measuring the distance to class-specific subspaces while training the embedding model instead of measuring the distance to a single or multiple centers as done for other angular margin losses.

Formally, the definition of the AdaProj loss is as follows.

**Definition 1** (AdaProj loss). Let $\mathcal{C}_k \subset \mathbb{R}^D$ with $|\mathcal{C}_k| = J \in \mathbb{N}$ denote class centers for class $k \in \{1, ..., N_{\text{classes}}\}$. Then for the AdaProj loss the logit for class $k \in \{1, ..., N_{\text{classes}}\}$ is defined as

$$L(x, \mathcal{C}_k) := \hat{s} \cdot \|P_{\mathcal{S}^{D-1}}(x) - P_{\mathcal{S}^{D-1}}(P_{\text{span}(\mathcal{C}_k)}(x))\|_2^2$$

where $\hat{s} \in \mathbb{R}_+$ is the so-called dynamically adaptive scale parameter of the AdaCos loss Zhang et al. (2019). Inserting these logits into a softmax function and computing the CCE yields the AdaProj loss function.

*Remark.* Note that, by Lemma 5 of Wilkinghoff & Kurth (2023), it holds that

$$\|P_{\mathcal{S}^{D-1}}(x) - P_{\mathcal{S}^{D-1}}(P_{\text{span}(\mathcal{C}_k)}(x))\|_2^2 = 2(1 - \langle P_{\mathcal{S}^{D-1}}(x), P_{\mathcal{S}^{D-1}}(P_{\text{span}(\mathcal{C}_k)}(x)) \rangle),$$

which is equal to the cosine distance in this case. This explains why the AdaProj loss can be called an angular margin loss.

As for other angular margin losses, projecting the embedding space onto the $D$-sphere has several advantages (Wilkinghoff & Kurth, 2023). Most importantly, if $D$ is sufficiently large randomly initialized centers are with very high probability approximately orthonormal to each other (Gorban et al., 2016), i.e. distributed equidistantly, and sufficiently far away from $\mathbf{0} \in \mathbb{R}^D$. Therefore, one does not need to carefully design a method to initialize the centers. Another advantage is that a normalization may prevent numerical issues, similar to applying batch normalization Ioffe & Szegedy (2015).

The following Lemma shows that using the AdaProj loss, as defined above, indeed allows the network to utilize a larger solution space.

**Lemma 2.** *Let $x \in \mathbb{R}^D$ and let $\mathcal{C} \subset \mathbb{R}^D$ contain pairwise orthonormal elements. Then,*

$$x \in \text{span}(\mathcal{C}) \cap \mathcal{S}^{D-1} \Rightarrow \|P_{\mathcal{S}^{D-1}}(x) - P_{\mathcal{S}^{D-1}}(P_{\text{span}(\mathcal{C})}(x))\|_2^2 = 0.$$

*Proof.* Let $x \in \text{span}(\mathcal{C}) \cap \mathcal{S}^{D-1} \subset \mathbb{R}^D$ with $|\mathcal{C}| = J$. Therefore, $\|x\|_2 = 1$ and there are $\lambda_j \in \mathbb{R}$ with $x = \sum_{j=1}^J \lambda_j c_j$. Thus, it holds that

$$x = \sum_{j=1}^J \lambda_j c_j = \sum_{j=1}^J \sum_{i=1}^J \lambda_i \langle c_i, c_j \rangle c_j = \sum_{j=1}^J \langle \sum_{i=1}^J \lambda_i c_i, c_j \rangle c_j = \sum_{j=1}^J \langle x, c_j \rangle c_j = P_C(x).$$

Hence,

$$\|P_C(x)\|_2 = \|x\|_2 = 1 \text{ as well as } \langle x, P_{\text{span}(\mathcal{C})}(x) \rangle = \langle x, x \rangle = \|x\|_2^2 = 1$$

and we obtain

$$
\begin{aligned}
&\|P_{\mathcal{S}^{D-1}}(x) - P_{\mathcal{S}^{D-1}}(P_{\mathrm{span}(\mathcal{C})}(x))\|_2^2 \\
=&\langle P_{\mathcal{S}^{D-1}}(x) - P_{\mathcal{S}^{D-1}}(P_{\mathrm{span}(\mathcal{C})}(x)), P_{\mathcal{S}^{D-1}}(x) - P_{\mathcal{S}^{D-1}}(P_{\mathrm{span}(\mathcal{C})}(x))\rangle \\
\\
=&\|P_{\mathcal{S}^{D-1}}(x)\|^2 - 2\langle P_{\mathcal{S}^{D-1}}(x), P_{\mathcal{S}^{D-1}}(P_{\mathrm{span}(\mathcal{C})}(x))\rangle + \|P_{\mathcal{S}^{D-1}}(P_{\mathrm{span}(\mathcal{C})}(x))\|^2 \\
=&1 - 2\frac{\langle x, P_{\mathrm{span}(\mathcal{C})}(x)\rangle}{\|x\|_2\|P_{\mathrm{span}(\mathcal{C})}(x)\|_2} + 1 \\
=&0.
\end{aligned}
$$

$\square$

*Remark.* If $\mathcal{C}$ contains randomly initialized elements of the unit sphere and $D$ is sufficiently large, then the elements of $\mathcal{C}$ are approximately pairwise orthonormal with very high probability (Gorban et al., 2016). Hence, this Lemma is likely to hold for the AdaProj loss.

When inserting the projection onto the $D-1$-sphere as an operation into the neural network, this Lemma shows that the solution space for the AdaProj loss function is increased to the whole subspace $\mathrm{span}(\mathcal{C})$, which has a dimension of $|\mathcal{C}|$ with very high probability. Because of this, it should be ensured that $|\mathcal{C}| < D$. Otherwise the whole embedding space may be an optimal solution and thus the network cannot learn a meaningful embedding space. In comparison, for the AdaCos loss only the class centers themselves are optimal solutions and for the sub-cluster AdaCos loss each sub-cluster is an optimal solution (Wilkinghoff & Kurth, 2023).

## 3 EXPERIMENTAL RESULTS

To experimentally evaluate the proposed AdaProj loss, first the experimental setup is presented by describing the datasets and the ASD system. After that, experimental results regarding the performance obtained with different loss functions, the impact of the subspace dimension and the relation to state-of-the-art systems are presented and discussed.

### 3.1 DATASETS AND PERFORMANCE METRICS

For the experiments, two ASD datasets, namely the DCASE2022 ASD dataset (Dohi et al., 2022a) and the DCASE2023 ASD dataset (Dohi et al., 2023) for semi-supervised machine condition monitoring, were used. Both datasets consist of a development set and an evaluation set that are divided into a training split containing only normal data and a test split containing normal as well as anomalous data. Furthermore, both tasks explicitly capture the problem of domain generalization (Wang et al., 2021) by defining a source and a target domain, which differs from the source domain by altering machine parameters or noise conditions. The task is to detect anomalous samples regardless of the domain a sample belongs to by training a system with only normal data. As meta information, the target machine type of each sample is known and for the training samples, also the domain and additional parameter settings or noise conditions, called attribute information, are known and thus can be utilized to train an embedding model.

The DCASE2022 ASD dataset (Dohi et al., 2022a) consists of the machine types "ToyCar" and "ToyTrain" from ToyAdmos2 (Harada et al., 2021) and "fan", "gearbox", "bearing", "slide rail" and "valve" from MIMII-DG (Dohi et al., 2022b). For each machine type, there are six different so-called sections, indicating different machine IDs of these types of which three belong to the development set and three belong to the test set. These IDs are known for each recording and can also be utilized as meta information to train the system. For the source domain of each section, there are 1000 normal audio recordings with a duration of $10\,\mathrm{s}$ with a sampling rate of $16\,\mathrm{kHz}$ belonging to the training split and 50 normal and 50 anomalous samples belonging to the test split. For the target domain of each section, there are also approximately 50 normal and 50 anomalous samples belonging to the test split but only 10 normal audio recordings belonging to the training split.

The DCASE2023 ASD dataset (Dohi et al., 2023) is similar to the DCASE2022 ASD dataset with the following modifications. First of all, the development set and the evaluation set contain mutually

Figure 1: Structure of the ASD system, adapted from Figure 1 in Wilkinghoff (2023b). Representation size in each step is given in brackets.

exclusive machine types. More concretely, the development set contains the same machine types as the DCASE2022 dataset and the evaluation set contains the machine types "ToyTank", "ToyNscale" and "ToyDrone" fromToyAdmos2+ (Harada et al., 2023b) and "vacuum", "bandsaw", "grinder" and "shaker" from MIMII-DG (Dohi et al., 2022b). Furthermore, there is only a single section for each machine type, which makes the auxiliary classification task much easier resulting in less informative embeddings for the ASD task. Last but not least, the duration of each recording has a length between $6\,\mathrm{s}$ and $18\,\mathrm{s}$. Overall, all three modifications make this task much more challenging than the DCASE2022 ASD task.

To measure the performance of the ASD systems the threshold-independent area under the receiver operating characteristic (ROC) curve (AUC) metric is used. In addition, the partial area under the ROC curve (pAUC) (McClish, 1989), which is the AUC for low false positive rates ranging from $0$ to $p = 0.1$ in this case, is used. The reason for incorporating the pAUC is that, for machine condition monitoring, one is interested in ensuring a low false alarm rate to not lose the trust of users in taking alarms seriously. Both performance metrics are computed domain-independent for every previously defined section of the dataset and the harmonic mean of all resulting values is the final score used to measure and compare the performances of different ASD systems.

## 3.2 ANOMALOUS SOUND DETECTION SYSTEM

For all experiments conducted in this work, the state-of-the-art ASD system presented in Wilkinghoff (2023a) is used. An overview of the system can be found in Figure 1. The system consists of three main components: 1) a feature extractor, 2) an embedding model and 3) a backend for computing anomaly scores.

In the first processing block, two different feature representations are extracted from the raw waveforms, namely magnitude spectrograms and the full magnitude spectrum. To make both feature representations a bit more different the temporal mean is subtracted from the magnitude spectrograms, essentially removing static frequency information that are captured with the highest possible resolution in the magnitude spectrums. Utilizing both of these representations was shown to significantly improve the performance (Wilkinghoff, 2023a) despite their close relation.

For each of the two feature representations, another convolutional subnetwork is trained and the resulting embeddings are concatenated and normalized with respect to the Euclidean norm to obtain a single embedding. In contrast to the original architecture, the embedding dimension is doubled from $256$ to $512$. More details about the subnetwork architectures can be found in Wilkinghoff (2023a). The network is trained for $10$ epochs using a batch size of $64$ using adam (Kingma & Ba, 2015) by utilizing meta information such as machine types and the provided attribute information as an auxiliary classification task. Different loss functions can be used for this purpose and will be compared in the next subsection. All loss functions investigated in this work require class-specific center vectors, which are initialized randomly using Glorot uniform intialization (Glorot & Bengio, 2010). To improve the resulting ASD performance, the randomly initialized class centers are not adapted during training and no bias terms are used as proposed in Ruff et al. (2018) for deep one-class classification. Furthermore, mixup (Zhang et al., 2018) with a uniformly distributed mixing coefficient is applied to the waveforms.

As a backend, k-means with 32 means is applied to the normal training samples of the source domain. For a given test sample, the smallest cosine distance to these means and the ten normal

Table 1: ASD performance obtained with different loss functions. Harmonic means of all AUCs and pAUCs over all pre-defined sections of the dataset are depicted in percent. Arithmetic mean and standard deviation of the results over ten independent trials are shown. Best results in each column are highlighted with bold letters.

| | source domain | | target domain | | domain-independent | |
|---|---|---|---|---|---|---|
| **DCASE2022 development set (Dohi et al., 2022a)** | | | | | | |
| loss function | AUC | pAUC | AUC | pAUC | AUC | pAUC |
| intra-class (IC) compactness loss (Ruff et al., 2018) | $81.8 \pm 1.6$ | $74.9 \pm 1.7$ | $75.3 \pm 1.0$ | $\mathbf{63.4 \pm 0.6}$ | $79.2 \pm 0.9$ | $64.7 \pm 1.1$ |
| IC compactness loss + CCE (Perera & Patel, 2019) | $82.5 \pm 1.8$ | $75.5 \pm 0.9$ | $75.5 \pm 0.7$ | $61.6 \pm 0.9$ | $79.0 \pm 0.8$ | $65.0 \pm 0.7$ |
| AdaCos loss (Zhang et al., 2019) | $82.6 \pm 1.4$ | $76.0 \pm 1.1$ | $76.5 \pm 1.2$ | $62.3 \pm 1.4$ | $79.8 \pm 0.7$ | $\mathbf{65.5 \pm 0.9}$ |
| sub-cluster AdaCos loss (Wilkinghoff, 2021) | $83.2 \pm 2.1$ | $75.9 \pm 1.3$ | $\mathbf{77.6 \pm 1.0}$ | $62.1 \pm 1.5$ | $80.0 \pm 1.4$ | $65.2 \pm 1.1$ |
| proposed AdaProj loss | $\mathbf{84.3 \pm 1.1}$ | $\mathbf{76.3 \pm 1.1}$ | $77.2 \pm 1.2$ | $62.2 \pm 1.1$ | $\mathbf{80.6 \pm 0.8}$ | $\mathbf{65.5 \pm 1.3}$ |
| **DCASE2022 evaluation set (Dohi et al., 2022a)** | | | | | | |
| loss function | AUC | pAUC | AUC | pAUC | AUC | pAUC |
| IC compactness loss (Ruff et al., 2018) | $74.7 \pm 0.9$ | $64.2 \pm 1.3$ | $65.9 \pm 0.8$ | $57.8 \pm 0.9$ | $70.3 \pm 0.8$ | $58.9 \pm 0.8$ |
| IC compactness loss + CCE (Perera & Patel, 2019) | $75.6 \pm 0.7$ | $66.9 \pm 0.8$ | $69.3 \pm 0.7$ | $59.3 \pm 0.7$ | $72.6 \pm 0.4$ | $60.3 \pm 0.7$ |
| AdaCos loss (Zhang et al., 2019) | $77.2 \pm 0.5$ | $65.9 \pm 1.4$ | $68.6 \pm 1.1$ | $58.6 \pm 0.7$ | $73.0 \pm 0.4$ | $59.7 \pm 0.6$ |
| sub-cluster AdaCos loss (Wilkinghoff, 2021) | $77.0 \pm 0.7$ | $66.5 \pm 0.9$ | $68.3 \pm 0.8$ | $58.8 \pm 0.6$ | $72.9 \pm 0.6$ | $59.5 \pm 0.5$ |
| proposed AdaProj loss | $\mathbf{77.4 \pm 1.0}$ | $\mathbf{67.0 \pm 0.6}$ | $\mathbf{69.7 \pm 0.6}$ | $\mathbf{59.6 \pm 0.6}$ | $\mathbf{73.6 \pm 0.7}$ | $\mathbf{60.5 \pm 0.7}$ |
| **DCASE2023 development set (Dohi et al., 2023)** | | | | | | |
| loss | AUC | pAUC | AUC | pAUC | AUC | pAUC |
| IC compactness loss (Ruff et al., 2018) | $67.0 \pm 2.1$ | $62.4 \pm 1.0$ | $69.1 \pm 1.4$ | $\mathbf{56.4 \pm 1.1}$ | $67.7 \pm 1.2$ | $56.9 \pm 0.9$ |
| IC compactness loss + CCE (Perera & Patel, 2019) | $70.6 \pm 1.8$ | $64.1 \pm 1.8$ | $71.2 \pm 1.4$ | $55.5 \pm 1.6$ | $70.4 \pm 1.0$ | $\mathbf{57.4 \pm 1.1}$ |
| AdaCos loss (Zhang et al., 2019) | $\mathbf{70.7 \pm 1.3}$ | $\mathbf{64.3 \pm 1.1}$ | $71.2 \pm 1.1$ | $55.4 \pm 1.3$ | $70.9 \pm 0.9$ | $56.8 \pm 0.9$ |
| sub-cluster AdaCos loss (Wilkinghoff, 2021) | $68.3 \pm 1.7$ | $62.0 \pm 1.5$ | $71.8 \pm 1.5$ | $55.6 \pm 1.5$ | $70.4 \pm 0.9$ | $56.3 \pm 0.8$ |
| proposed AdaProj loss | $70.3 \pm 1.7$ | $61.8 \pm 1.6$ | $\mathbf{72.2 \pm 1.4}$ | $55.1 \pm 1.1$ | $\mathbf{71.4 \pm 1.0}$ | $56.2 \pm 0.7$ |
| **DCASE2023 evaluation set (Dohi et al., 2023)** | | | | | | |
| loss | AUC | pAUC | AUC | pAUC | AUC | pAUC |
| IC compactness loss (Ruff et al., 2018) | $73.5 \pm 1.8$ | $63.4 \pm 1.8$ | $58.8 \pm 2.5$ | $55.7 \pm 1.3$ | $64.0 \pm 1.5$ | $55.8 \pm 0.9$ |
| IC compactness loss + CCE (Perera & Patel, 2019) | $74.3 \pm 1.5$ | $\mathbf{64.0 \pm 1.6}$ | $61.6 \pm 2.0$ | $55.7 \pm 0.9$ | $67.5 \pm 0.8$ | $57.5 \pm 1.0$ |
| AdaCos loss (Zhang et al., 2019) | $\mathbf{74.7 \pm 1.5}$ | $63.8 \pm 1.8$ | $61.6 \pm 3.4$ | $57.1 \pm 1.4$ | $68.0 \pm 1.6$ | $58.0 \pm 1.1$ |
| sub-cluster AdaCos loss (Wilkinghoff, 2021) | $73.2 \pm 1.9$ | $61.6 \pm 1.4$ | $62.0 \pm 2.2$ | $55.8 \pm 1.3$ | $66.5 \pm 1.6$ | $56.2 \pm 1.0$ |
| proposed AdaProj loss | $74.2 \pm 1.8$ | $62.9 \pm 1.0$ | $\mathbf{64.4 \pm 2.0}$ | $\mathbf{57.7 \pm 0.8}$ | $\mathbf{69.8 \pm 1.3}$ | $\mathbf{60.0 \pm 0.5}$ |
| **arithmetic mean over all datasets** | | | | | | |
| loss | AUC | pAUC | AUC | pAUC | AUC | pAUC |
| IC compactness loss (Ruff et al., 2018) | 74.3 | 66.2 | 67.3 | 58.3 | 70.3 | 59.1 |
| IC compactness loss + CCE (Perera & Patel, 2019) | 75.8 | **67.6** | 69.4 | 58.0 | 72.4 | 60.1 |
| AdaCos loss (Zhang et al., 2019) | 76.3 | 67.5 | 69.5 | 58.4 | 72.9 | 60.0 |
| sub-cluster AdaCos loss (Wilkinghoff, 2021) | 75.4 | 66.5 | 69.9 | 58.1 | 72.5 | 59.3 |
| proposed AdaProj loss | **76.6** | 66.3 | **70.9** | **58.7** | **73.9** | **60.6** |

training samples of the target domain is used as an anomaly score. Thus, smaller values indicate normal samples whereas higher values indicate anomalous samples.

## 3.3 PERFORMANCE EVALUATION

The first and most important experiment is to compare different loss functions for training the embedding extractor of the ASD system presented in the previous subsection. For this purpose, we used 1) individual class-specific IC compactness losses jointly trained on all classes, as proposed for one-class classification in Ruff et al. (2018), 2) an additional discriminative CCE loss, similar to the descriptiveness loss used in Perera & Patel (2019) but trained on the same dataset, 3) the AdaCos loss (Zhang et al., 2019), 4) the sub-cluster AdaCos loss (Wilkinghoff, 2021) with 32 sub-clusters and 5) the proposed AdaProj loss. The experiments were conducted on the development and evaluation split of the DCASE2022 and the DCASE2023 ASD dataset and each experiment was repeated ten times to reduce the variance of the resulting performances. Furthermore, the arithmetic means of the performances obtained on the different datasets are shown to be able to directly compare the overall performance. The results can be found in Table 1.

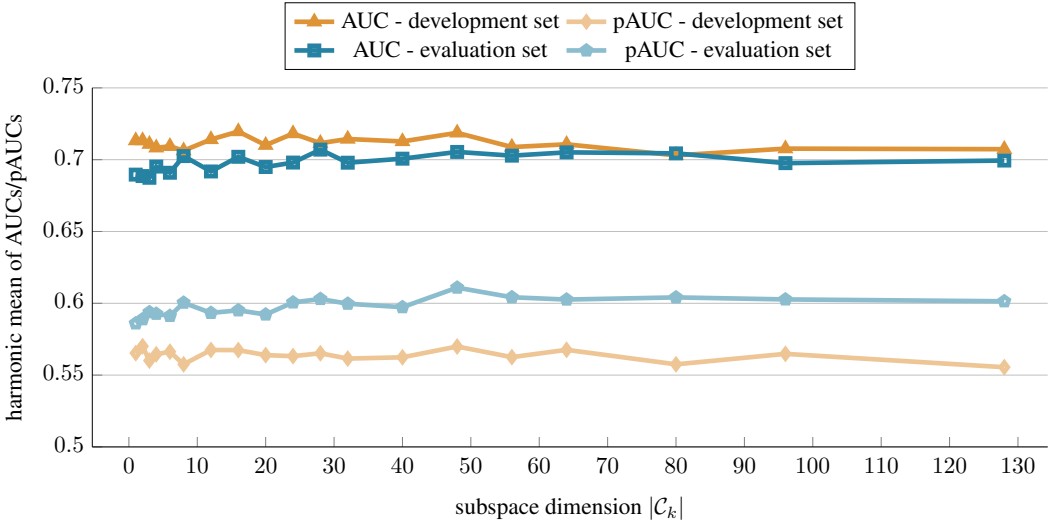

Figure 2: Domain-independent performance obtained on the DCASE2023 dataset with different subspace dimensions. The means over ten independent trials are shown.

The main observation to be made is that the proposed AdaProj loss clearly outperforms all other losses on both datasets. Especially on the DCASE2023 dataset, there are significant improvements to be observed. The most likely explanation is that for this dataset the classification task is less difficult and thus a few classes may be easily identified leading to embeddings that do not carry enough information to distinguish between embeddings belonging to normal and anomalous samples of these classes.

Another interesting observation is that, in contrast to the original results presented in Wilkinghoff (2021), the sub-cluster AdaCos loss actually performs slightly worse than the AdaCos loss despite having a higher solution space. A possible explanation is that in this work, the centers are adapted during training whereas, in our work, they are not as this has been shown to improve the resulting performance (Wilkinghoff, 2023a). Since all centers have approximately the same distance to each other when being randomly initialized, i.e. the centers belonging to a target class and the other centers, the network will likely utilize only a single center for each class that is closest to the initial embeddings of the corresponding target class. Moreover, a low inter-class similarity is more difficult to ensure due to the higher total number of sub-clusters belonging to other classes. This leads to more restrictive requirements when learning class-specific distributions and thus actually reduces the ability to differentiate between embeddings belonging to normal and anomalous samples.

### 3.4 INVESTIGATING THE IMPACT OF THE SUBSPACE DIMENSION ON THE PERFORMANCE

As an ablation study, different choices for the dimension of the subspaces have been compared experimentally on the DCASE2023 ASD dataset. The results can be found in Figure 2. It can be seen, that, on the development set, the results are relatively stable while a larger dimension slightly improves the performance on the evaluation set without any significant differences. For subspace dimensions greater than 48 the performances seem to slightly degrade again. In conclusion, the subspace dimension should be neither too high nor too low and a dimension of 32 as used for the other experiments in this works appears to be a reasonable choice.

### 3.5 COMPARISON TO OTHER PUBLISHED SYSTEMS

As a last experiment, the performance of the proposed system using AdaProj is compared to the ten top-performing systems of the DCASE2023 Challenge. As many systems utilize ensembles of models, the mean of the anomaly scores belonging to ten independent trials was used to create an ensemble of ten systems allowing a fair comparison. The results can be found in Figure 3. It can

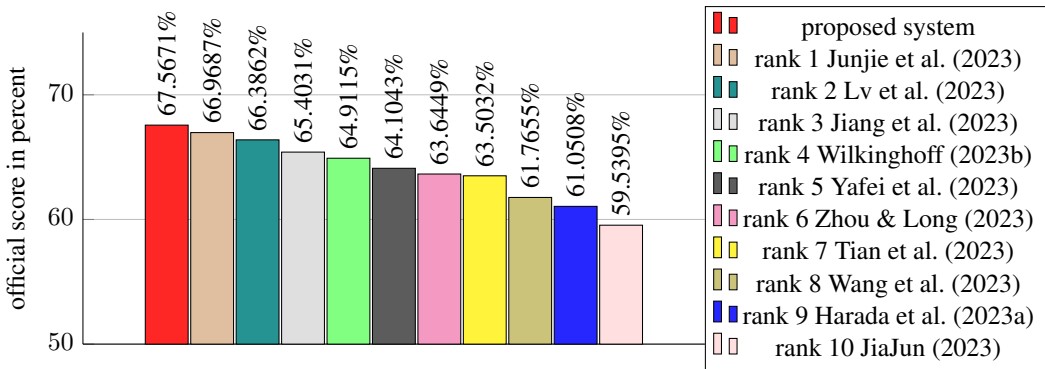

Figure 3: Comparison of the proposed system to the ten top-performing systems of the DCASE2023 Challenge. The official evaluation script was used to compute the score.

be seen that the proposed system outperforms all other published systems and thus achieves a new state-of-the-art performance. This adds confidence to the benefits of the AdaProj loss function.

## 4 CONCLUSIONS AND FUTURE WORK

In this work, AdaProj a novel angular margin loss function specifically designed for semi-supervised anomaly detection with auxiliary classification tasks was presented. It was proven that this loss function learns an embedding space with class-specific subspaces of arbitrary dimension. In contrast to other angular margin losses, which try to project data to individual points in space, this relaxes the requirements of solving the classification task and allows for less compact distributions in the embedding space. In experiments conducted on the DCASE2022 and DCASE2023 ASD datasets, it was shown that using AdaProj results in better performance than other commonly used loss functions. In conclusion, the resulting embedding space has a more desirable structure than the other embedding spaces for differentiating between normal and anomalous samples. As a result, a new state-of-the-art performance outperforming all other published systems could be achieved on the DCASE2023 ASD dataset. For future work, it is planned to evaluate AdaProj on other datasets and using other auxiliary classification tasks, e.g. tasks imposed by SSL.

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
