# OpenReview forum: "AdaProj: Adaptively Scaled Angular Margin Subspace Projections for Anomaly Detection with Auxiliary Classification Tasks"
_ICLR.cc/2024/Conference — ICLR 2024 Conference Withdrawn Submission_

### Official Review · Reviewer_3pzd · 2023-10-13

**Soundness:** 2 fair
**Presentation:** 2 fair
**Contribution:** 1 poor
**Rating:** 3
**Confidence:** 4

**Summary:**

The paper proposes a new loss function for an outlier detection method based on adacos. The idea is to train an embedding space for an auxiliary classification task. The classifier learns class centroids and the distance to these centroids represent the logits fed into the cross entropy loss. In AdaCos, the loss is computed with respect to the cosine distance between the class centroid and the vector. In AdaProj, the authors propose to use the Euclidian distance of projections on the unit hypersphere instead.
The loss is evaluated within an existing outlier detection pipline on anomalous sound detection challenge data sets from the last year. The prposed method shows an improvement even though the gap is not extreme.

**Strengths:**

The paper is clearly written even though some of the similarities to early works are not distinguished very clearly. The paper improves the state of the art with respect to the  DCASE challenge of 22 and 23 and demonstrates that the loss function can improve the performance on averge.

**Weaknesses:**

The paper's contribution is rather limited as mostly replaces the consine similarity with the very similar concept of the L2 distance of projections to the unit sphere. The justification of this step where rather cryptic and I could not distinguish the argument of AdaProj over AdaCos from arguments for the whole design of centroid-based embeddings for outlier detection. However, this was mostly proposed in already published works.
The impact  of the lemma to the proposed methond seems  also remained unclear to me.
The experimental evaluation and most part of the paper seem to be directed to improve thes DCASE challenge. Thus, the improvement might be interesting for this particular community. But in order to demonstrate that the proposed loss function is of general interest to the broader AI community, a more widespride set field of applications might be necessary.

To conclude, the contribution seems not very broad and the motivation of why it is a recognizable contribution to a broader stater-of-the-art in representation learning is insufficient.

**Questions:**

In lemma 2, you assume that x is part of the unit sphere as well as the subspace. Wouldn't this imply that the projection into both spaces must be x itself ?

---

### Official Review · Reviewer_RkC3 · 2023-10-16

**Soundness:** 2 fair
**Presentation:** 1 poor
**Contribution:** 2 fair
**Rating:** 3
**Confidence:** 4

**Summary:**

The paper addresses the challenge of semi-supervised sound anomaly detection, a critical task in machine condition monitoring. The authors introduce AdaProj, which utilizes an angular margin loss function. This loss function learns to project data onto class-specific subspaces rather than aiming to bring data as close as possible to their respective class centers.

**Strengths:**

AdaPoj, the proposed method, demonstrates state-of-the-art performance as demonstrated by rigorous empirical evaluations on the DCASE2022 and DCASE2023 datasets.

**Weaknesses:**

1. The paper exhibits a notable lack of clarity in its presentation. The problem statement and underlying setting are not adequately explained, necessitating multiple readings to grasp the details. For instance, there is ambiguity concerning class definition within this context. Given that anomaly detection traditionally aligns with a one-class classification paradigm, the introduction of classes in this approach requires a more thorough explanation. Additionally, the distinction between datasets and their respective splits remains unclear. A clear definition of the distinctions between evaluation and development sets is crucial for comprehensive understanding. Presently, the manuscript falls short of readiness and would greatly benefit from substantial revisions, potentially utilizing the available spare pages to elaborate upon its nuances, especially given that the paper is 7.5 pages in length, well below the 9-page limit.

2. My fundamental concern centers on the degree of incremental novelty this work contributes to. Specifically, the entire framework is constructed upon an existing ASD system introduced in past works. The primary alteration appears to be a subtle adjustment to the auxiliary objective. While the proposed AdaProj approach represents an intriguing extension, there is a need for a more detailed discussion regarding the substantial departure or enhancement it brings to the existing framework.

**Questions:**

See weaknesses.

---

### Official Review · Reviewer_t49Z · 2023-10-30

**Soundness:** 2 fair
**Presentation:** 2 fair
**Contribution:** 2 fair
**Rating:** 6
**Confidence:** 4

**Summary:**

The paper appears to focus on a novel loss function called the AdaProj loss, which is designed to improve the performance of anomaly sound detection (ASD) systems. The AdaProj loss aims to enlarge the space of optimal solutions, allowing the network to learn less restrictive distributions of normal samples. This is expected to help differentiate between normal and anomalous data after training. The AdaProj loss measures the distance to class-specific subspaces during the training of the embedding model, rather than measuring the distance to a single or multiple centers as done for other angular margin losses. The paper compares the AdaProj loss with other loss functions, such as the AdaCos loss and sub-cluster AdaCos loss, on datasets like DCASE2022 and DCASE2023. The results suggest that the proposed AdaProj loss outperforms other losses, especially on the DCASE2023 dataset.

**Strengths:**

1. The AdaProj loss introduces a new approach to learning embeddings for ASD systems, focusing on class-specific linear subspaces rather than single or multiple centers.
2. The AdaProj loss demonstrates superior performance compared to other loss functions, especially on the DCASE2023 dataset.
3. The paper provides lemmas and proofs to support the claims made about the AdaProj loss, adding depth to the research.
4. The paper conducts a thorough comparison of the AdaProj loss with other existing loss functions, providing a comprehensive view of its advantages.

**Weaknesses:**

1. The paper seems to delve deep into mathematical formulations, which might make it challenging for readers without a strong mathematical background.
2. The extracted summary does not provide a clear context or background on the significance of the problem being addressed, which might make it difficult for readers unfamiliar with ASD systems to grasp the paper's importance.
3. The paper primarily focuses on the DCASE2022 and DCASE2023 datasets. Including more diverse datasets could have provided a more comprehensive evaluation.
4. The summary does not mention any details about the network architectures or other hyperparameters used, which might be crucial for reproducibility.

**Questions:**

See weaknesses.

---

### Official Review · Reviewer_PZSP · 2023-10-31

**Soundness:** 2 fair
**Presentation:** 2 fair
**Contribution:** 2 fair
**Rating:** 5
**Confidence:** 3

**Summary:**

This paper proposes a new loss term for semi-supervised anomaly detection. It is designed to learn a class-specific subspace to facilitate the detection process. Experiments are conducted on two datasets and the proposed loss function achieves the SOTA results on DCASE2023 dataset.

**Strengths:**

1. Semi-supervised anomaly detection is typical yet valuable topic for general machine learning research. This paper keeps exploring this direction.
2. Experiments on two datasets show the proposed loss outperforms other baselines.

**Weaknesses:**

1. Comparison methods are not recently published works, adding more recent one or two years publications helps to further support the loss effectiveness.
2. The empirical results may need more analysis and discussion. Why the proposed loss works better than others? simply comparing the numerical performance cannot provide more intuition for a newly designed term.
3. The whole draft needs a revision to be more informative. For example, the anomaly detection may need a figure illustration to show the detection results with corresponding discussion.
4. Experiments on two datasets may not sufficient and the results are not consistently better than other baselines.

**Questions:**

Please refer to the weakness for details.